# Highly Oxygenated Triterpenoids and Diterpenoids from Fructus Rubi (*Rubus chingii* Hu) and Their NF-kappa B Inhibitory Effects

**DOI:** 10.3390/molecules26071911

**Published:** 2021-03-29

**Authors:** Jiang Wan, Xiao-Juan Wang, Nan Guo, Xi-Ying Wu, Juan Xiong, Yi Zang, Chun-Xiao Jiang, Bing Han, Jia Li, Jin-Feng Hu

**Affiliations:** 1Minhang Hospital & School of Pharmacy, Fudan University, Shanghai 201199, China; 18111030021@fudan.edu.cn (J.W.); 18221765046@163.com (X.-J.W.); gnguoguo@163.com (N.G.); wxyifd@163.com (X.-Y.W.); jxiong@fudan.edu.cn (J.X.); 2State Key Laboratory of Drug Research, Shanghai Institute of Materia Medica, Chinese Academy of Science, Shanghai 201203, China; yzang@simm.ac.cn; 3Institute of Natural Medicine and Health Products, School of Advance Study, Zhejiang Provincial Key Laboratory of Plant Ecology and Conservation, Taizhou University, Taizhou 318000, China; jiangchx15@tzc.edu.cn

**Keywords:** *Rubus chingii* Hu, Rosaceae, triterpenoids, rubusacids, diterpenoids, rubusone, NF-*κ*B

## Abstract

During a phytochemical investigation of the unripe fruits of *Rubus chingii* Hu (i.e., Fructus Rubi, a traditional Chinese medicine named “Fu-Pen-Zi”), a number of highly oxygenated terpenoids were isolated and characterized. These included nine ursane-type (**1**, **2**, and **4**–**10**), five oleanane-type (**3**, **11**–**14**), and six cucurbitane-type (**15**–**20**) triterpenoids, together with five *ent*-kaurane-type diterpenoids (**21**–**25**). Among them, (4*R*,5*R*,8*R*,9*R*,10*R*,14*S*,17*S*,18*S*,19*R*,20*R*)-2,19α,23-trihydroxy-3-oxo-urs-1,12-dien-28-oic acid (rubusacid A, **1**), (2*R**,4*S**,5*R**,8*R**,9*R**,10*R**,14*S**,17*S**, 18*S**,19*R**,20*R**)-2α,19α,24-trihydroxy-3-oxo-urs-12-en-28-oic acid (rubusacid B, **2**), (5*R*,8*R*,9*R*,10*R*, 14*S*,17*R*,18*S*,19*S*)-2,19α-dihydroxy-olean-1,12-dien-28-oic acid (rubusacid C, **3**), and (3*S*,5*S*,8*S*,9*R*, 10*S*,13*R*,16*R*)-3α,16α,17-trihydroxy-*ent*-kaur-2-one (rubusone, **21**) were previously undescribed. Their chemical structures and absolute configurations were elucidated on the basis of spectroscopic data and electronic circular dichroism (ECD) analyses. Compounds **1** and **3** are rare naturally occurring pentacyclic triterpenoids featuring a special *α,β*-unsaturated keto-enol (diosphenol) unit in ring A. Cucurbitacin B (**15**), cucurbitacin D (**16**), and 3*α*,16*α*,20(*R*),25-tetrahydroxy-cucurbita-5,23- dien-2,11,22-trione (**17**) were found to have remarkable inhibitory effects against NF-*κ*B, with IC_50_ values of 0.08, 0.61, and 1.60 μM, respectively.

## 1. Introduction

The *Rubus* genus (family Rosaceae) is large and diverse (with about 700 species distributed worldwide), and Flora of China lists 139 species as endemic to China [1]. They are usually deciduous or semi-evergreen perennial herbs or shrubs, which are often spiny, with a characteristic fruit formed as a head of one-seeded drupelets. The unripe fruits of *Rubus chingii* Hu (Fructus Rubi, referred to as “Fu-Pen-Zi” in Chinese, bog-bun-ja in Korean, and gosho-ichigo in Japanese) have been widely used as a herb tonic for the treatment of various diseases, mainly associated with kidney deficiency, in East Asian countries [2,3]. As a top-grade traditional Chinese medicine, Fructus Rubi was recorded in one of the earliest collections in the Pharmacopeia of the People’s Republic of China [4]. The phytochemistry and pharmacology of *R. chingii* have recently been well*-*documented by two review articles [2,5], and both triterpenoids and diterpenoids are encountered in the fruits. Modern pharmacological studies have revealed that the chemical components from Fructus Rubi exhibit a broad spectrum of bioactivities, such as being anti-aging [6], anti-cancer [7], anti-oxidant [7,8], and anti-diabetic [9]. With the aim of obtaining more structurally interesting and bioactive naturally occurring triterpenoids and diterpenoids [10,11,12], a phytochemical investigation of a commercially available sample of Fructus Rubi was carried out, which resulted in the isolation of three new highly oxygenated pentacyclic triterpenoids (**1**–**3**) and one new *ent*-kaurane-type diterpenoid (**21**), along with 21 related known terpenoid compounds (**4**–**20** and **22**–**25**) (Figure 1). Herein, the isolation, structural elucidation, and NF-*κ*B inhibitory activities of these compounds are reported.

## 2. Results and Discussion

A 70% ethanol extract of the unripe fruits of *R. chingii* (28.0 kg) was suspended in H_2_O and then partitioned successively with petroleum ether, EtOAc, and *n*-BuOH. The entire EtOAc-soluble fraction was repeatedly subjected to column chromatography (CC) over silica gel, MCI gel, Sephadex LH-20, and semi-preparative HPLC to afford 20 triterpenoids (**1**–**20**) and 5 diterpenoids (**21**–**25**) (Figure 1). By comparing the observed and reported physicochemical properties and spectroscopic data, the previously known ones were identified as fupenzic acid (**4**) [13], 2*α*,3*α*,19*α*-trihydroxy-urs-12-en-28-oic acid (**5**) [14], 2*α*,3*α*,23-trihydroxy-urs-12-en-28-oic acid (**6**) [15], 2*α*,19*α*-dihydroxy-3-oxo-urs-12-en-28-oic acid (**7**) [16], 2*α*,3*β*,19*α*,24-tetrahydroxyurs-12-en-28-oic acid (**8**) [17], 1*β*,3*β*,19*α*-trihydroxy-2-oxo-urs-12-en-28-oic acid (**9**) [18], 3*β*,19*α*-dihydroxy-2-oxo-urs-12-en-28-oic acid (**10**) [19], 2*α*,3*α*,19*α*,23-tetrahydroxy-olean-12-en-28-oic acid (**11**) [20], 2*α*,3*α*,23-trihydroxy-urs-12-en-28-oic acid (**12**) [21], arjunic acid (**13**) [22], 2*α*,3*β*,19*α*,24-tetrahydroxyolean-12-en-28-oic acid (**14**) [23], cucurbitacin B (**15**) [24], cucurbitacin D (**16**) [25], 3*α*,16*α*,20(*R*),25-tetrahydroxy-cucurbita-5,23-dien-2,11,22-trione (**17**) [26], 2,16*α*,20(*R*),25-tetrahydroxy-cucurbita-1,5,23-trien-3,11,22-trione (**18**) [27], 25-acetoxy-2*α*,16*α*,20(*R*)-trihydroxy-cucurbita-5,23-dien-3,11,22-trione (**19**) [28], 25-acetoxy-3*β*,16*α*,20(*R*)-trihydroxy-cucurbita-5,23-dien-2,11,22-trione (**20**) [26], 3*β*,16*α*,17-trihydroxy-*ent*-kaur-19-yl acetate (**22**) [29], 16*α*,17-dihydroxy-*ent*-kaur-3-one (**23**) [30], 3*β*,16*α*,17-trihydroxy-*ent*-kaurane (**24**) [31], and 16*α*,17,19-trihydroxy-*ent*-kaur-3-one (**25**) [32], respectively.

Rubusacid A (**1**) was obtained as a white powder. Its molecular formula was established as C_30_H_44_O_6_ from its HRESIMS (*m*/*z* 499.3067 [M − H]^−^, calcd. for C_30_H_43_O_6_, 499.3065) and ^13^C NMR data (Table 1). The IR spectrum exhibited characteristic absorptions for a cyclic enone (1703, 1644 cm^−1^) group, which was supported by its UV absorption band at 265 nm [33,34]. In the upfield region of the ^1^H NMR spectrum of **1**, resonances of five tertiary methyl groups at *δ*_H_ 0.90 (3H, s, Me-26), 1.23 (3H, s, Me-29), 1.29 (3H, s, Me-24), 1.30 (3H, s, Me-25), and 1.36 (3H, s, Me-27), and one secondary methyl group at *δ*_H_ 0.95 (3H, d, *J* = 7.1 Hz, Me-30) were observed (Table 2). In addition, signals resonating at *δ*_H_ 3.69 and 3.71 (ABq, each 1H, d, *J* = 12.0 Hz, H_2_-23) for a hydroxymethylene group, and two olefinic protons at *δ*_H_ 5.37 (1H, dd, *J* = 3.7, 3.5 Hz, H-12) and 6.29 (1H, s, H-1) were also readily distinguished. A total of 30 carbon signals, including a ketone carbonyl at *δ*_C_ 200.5 (C-3), a carboxyl carbon at *δ*_C_ 182.2 (C-28), four olefinic carbons at *δ*_C_ 145.8 (C-2), 140.4 (C-13), 129.5 (C-1), and 129.0 (C-12), and two oxygenated carbons at *δ*_C_ 73.5 (C-19) and 65.9 (C-23), were displayed in its ^13^C NMR spectrum. The aforementioned NMR data of **1** highly resembled those of fupenzic acid (**4**) [13], a co-occurring ursane-type triterpenoid with an *α*,*β*-unsaturated keto-enol moiety (i.e., diosphenol chromophore [13,35,36]) in ring A. The only difference between them was that the Me-23 in **4** was hydroxylated in **1**, which was confirmed by the H_3_-24/C-23 and H_2_-23/C-3 correlations (Figure 2) in its HMBC NMR spectrum. The relative configuration of **1** was determined by ROESY data analysis (Figure 3). The ROE correlation of H_2_-23 with H-5, along with the absence between H_2_-23 and H_3_-25, confirmed the *α*-orientation of the 23-CH_2_OH group. Moreover, the ROE correlations of H-18 with H-12/H-20/H_3_-29, and of H_3_-29 with H-12/H-20 demonstrated that H-18, H-20, and H_3_-29 were all *β*-oriented, thus requiring OH-19 in the *α*-orientation. The electronic circular dichroism (ECD) spectrum of **1** exhibited Cotton effects (CEs) at 217 (Δε −16.3), 274 (Δε +13.9), and 330 (Δε −3.0) nm (Appendix A, Supporting Information), corresponding to the *n*→*π** and *π*→*π** transitions of the unsaturated cyclohexanone chromophore. The ECD data of **1** were similar to those of (+)-cyrillin A [34] and sapiumic acid F [37], two pentacyclic triterpenoids possessing a common enone group in ring A, which then allowed the assignment of a 10*R* configuration in **1**. Consequently, the structure of **1** was characterized as (4*R*,5*R*,8*R*,9*R*,10*R*,14*S*,17*S*,18*S*,19*R*,20*R*)-2,19*α*,23-trihydroxy-3-oxo- urs-1,12-dien-28-oic acid.

The HRESIMS data permitted the molecular formula of rubusacid B (**2**) to be assigned as C_30_H_46_O_6_ (*m*/*z* 525.3184 [M + Na]^+^, calcd. for C_30_H_46_O_6_Na, 525.3187), which was consistent with its ^13^C NMR data (Table 1). The close similarity of the ^1^H (Table 2) and ^13^C NMR spectroscopic data of **2** with those of **1**, indicated that **2** is also an ursane-type triterpenoid. The difference of two mass units between **1** and **2**, and the absence of the enol group (*δ*_H_ 6.29, s (H-1); *δ*_C_ 129.5 (C-1), 145.8 (C-2)) in ring A when compared with those of **1**, suggested that the Δ^1^ double bond in **1** was hydrogenated in **2**. This was confirmed by the HMBC correlation from H_3_-25 to C-1 (*δ*_C_ 51.2) (Figure 2). The large coupling constant of *J*_1*α*,2_ = 12.7 Hz suggested the axial position for H-2. Moreover, the correlations of H-2/H_3_-25, H-2/H_2_-24, and H_2_-24/H_3_-25 in the ROESY NMR experiment (Figure 3) confirmed the *β*-orientation for both H-2 and the CH_2_OH group at C-4. In addition, the *α*-orientation of OH-19 was also determined by ROESY NMR experiment, in a similar way to **1** (Figure 3). Accordingly, the structure of **2** was deduced as (2*R**,4*S**,5*R**,8*R**,9*R**,10*R**,14*S**,17*S**,18*S**,19*R**,20*R**)-2*α*,19*α*,24-trihydroxy-3-oxo-urs-12-en-28-oic acid.

Rubusacid C (**3**) had a molecular formula of C_30_H_44_O_5_, as determined by HRESIMS ([M + H]^+^ ion at *m*/*z* 485.3254, calcd for C_30_H_45_O_5_, 485.3262) and ^13^C NMR data (Table 1). Like **1**, the IR (1695, 1634 cm^−1^) and UV (266 nm) absorption bands of **3** also indicated a cyclic enonic group. The ^1^H (Table 2) and ^13^C NMR data of **3** showed similarities to those of **1**, with the obvious differences being the presence of a pair of germinal methyl groups at C-20 (*δ*_H_ 0.97, s; 0.94, s) and an oxygenated methine at C-19 (*δ*_H_ 3.26, d, *J* = 3.9 Hz; *δ*_C_ 82.4) in **3**, rather than the secondary methyl group and tertiary methyl group in **1**. This implied that **3** is an oleanane-type triterpenoid, which agreed with the co-occurring terpenoid compounds **11**–**14** [20,21,22,23]. This was further supported by HMBC correlations (Figure 2) from H_3_-29 and H_3_-30 to C-19, C-20, and C-21, and from H-18 to C-19. In addition, the OH-23 in **1** was absent in **3**, which was corroborated by the HMBC cross-peaks from H_3_-24 to C-23. The relative configuration of **3** was determined by analyzing the coupling constants and ROESY data. The small *J* value (3.9 Hz) between H-18 and H-19 was indicative of the equatorial orientation of H-19. Moreover, the ROE correlations from H-19 to H-12, H-18, and H_3_-29 confirmed that OH-19 was *α*-oriented (Figure 3). The chirality of C-10 was assigned to be *R*, as evidenced from a positive CE at 274 nm and negative CEs at 215 and 333 nm in its ECD spectrum, which were comparable with (+)-cyrillins A and relevant analogues [34,37]. Therefore, the structure of **3** was defined as (5*R*,8*R*,9*R*,10*R*,14*S*,17*R*,18*S*,19*S*)-2,19*α*-dihydroxy-olean-1,12-dien-28-oic acid.

Rubusone (**21**) was obtained as a white, amorphous powder. It had a molecular formula of C_20_H_32_O_4_ according to HRESIMS data analysis ([M + H]^+^
*m*/*z* 337.2389, calcd for C_20_H_33_O_4_, 337.2373) and the ^13^C NMR data (Table 1). The IR absorption bands at 3379 and 1701 cm^−1^ suggested the presence of hydroxyl and carbonyl groups, respectively. Inspection of the ^1^H NMR spectroscopic data (in C_6_D_6_, Table 3) indicated the presence of three tertiary methyl groups (*δ*_H_ 0.67 (3H, s, Me-19), 0.69 (3H, s, Me-20), and 1.10 (3H, s, Me-18)), an oxymethine resonance (*δ*_H_ 3.62 (1H, d, *J* = 3.8 Hz, H-3)), and a hydroxymethylene group at *δ*_H_ 3.37 and 3.44 (ABq, each 1H, d, *J* = 10.3 Hz, H_2_-17). A total of 20 carbon signals, including one ketone carbonyl at *δ*_C_ 210.2 (C-2) and three oxygenated at *δ*_C_ 82.8 (C-3), 81.2 (C-16), and 66.3 (C-17), were displayed in its ^13^C NMR spectrum. The aforementioned NMR data were similar to those of the co-occurring compounds **22**–**25** [29,30,31,32], suggesting **4** is an *ent*-kaurane derivative. Detailed comparisons suggested that the 1D NMR data were closely related to those of 3*α*,16*β*,17-trihydroxy-*ent*-kaurane-2-one, which was previously isolated from *Homalanthus acuminatus* [38]. The only noticeable difference was that the chemical shifts of CH_2_OH-17 (*δ*_H_: 3.37 and 3.44, *δ*_C_: 66.3) in **21** were significantly distinguished when compared with 3*α*,16*β*,17-trihydroxy-*ent*-kaurane-2-one (*δ*_H_: 3.09 and 3.12; *δ*_C_: 69.8), indicating that these two compounds are epimeric at C-16. The *α*-orientation of OH-16, as with **22**–**25** [29,30,31,32], was confirmed by comparing the chemical shifts of C-16 and C-17 with reported data [31,39]. The magnitude of *J*_H-5/H-6_*_α_* (11.6 Hz) indicated that H-5 was in axial position, and the ROE correlations of H-5 with H-3/H-9, and of H-3 with H_3_-18 revealed their cofacial relationship (arbitrarily assigned as *β*-oriented). In turn, the ROE correlation of H_3_-19 and H_3_-20 was indicative of their *α*-orientation. Additionally, the ECD spectrum of **21** showed a negative CE at 280 nm arising from the *n*→*π** transition of the C-2 carbonyl group, indicative of an *S* configuration for C-3 by using the octant rule [40,41]. Thus, the structure of **21** was elucidated as (3*S*,5*S*,8*S*,9*R*,10*S*,13*R*,16*R*)-3*α*,16*α*,17-trihydroxy-*ent*-kaur-2-one.

Nuclear factor-*κ*B (NF-*κ*B) is an important transcription factor controlling different biological processes, such as immune differentiation and activation [42]. It has been regarded as a potential target for the regulation of dysfunction of immunity and inflammation. Recently, a few triterpenoids and diterpenoids from *Stewartia sinensis* [11] and *Pseudotsuga sinensis* [12] were found to have significant NF-*κ*B inhibitory effects. In the present study, all the isolated terpenoids (**1**–**25**) were evaluated for their NF-*κ*B inhibitory activities. Among them, only three cucurbitane-type compounds, **15**–**17**, showed potent inhibitory effects, with IC_50_ values of 0.08, 0.61, and 1.60 μM (Table 4), respectively. The rest of the isolates were inactive (inhibition ration < 50% at 20 μM). Bortezomib (PS-341) was used as the positive control (IC_50_: 0.44 μM) [43]. Interestingly, some *ent*-labdane-type diterpenoid glycosides from Fructus Rubi have been reported to suppress the NF-*κ*B signaling pathway [44], but the *ent*-kaurane-type diterpenoids isolated herein were inactive against NF-*κ*B.

In addition, the 70% ethanol extract of Fructus Rubi combined with fluconazole (FLC) has been previously found to have an anti-fungal activity against twenty-two FLC-resistant *Candida albicans* strains [45]. Hence, all the isolated compounds were subjected to the same bioassay, but none of them were active. They were also evaluated for their anti-fungal effects against the *C. albicans* SC5314 sensitive strain, and only compound **12** showed a moderate inhibition (MIC_80_: 32 μg/mL), whereas the other twenty-four isolates were inactive (MIC_80_ > 64 μg/ mL). Fluconazole (MIC_80_: 0.125 μg/mL) was used as the positive control [46].

## 3. Materials and Methods

### 3.1. General Experimental Procedures and Agents

Optical rotations were obtained with a Rudolf Autopol IV at 21 °C. UV and IR spectra were recorded on a Hitachi U-2900E UV spectrophotometer (Hitachi High-Techologies Corporation, Tokyo, Japan) and a Thermo Scientific Nicolet Is5 FT-IR spectrometer (Thermo Fisher Scientific, San Jose, CA, USA), respectively. ECD spectra were collected on a JASCO-810 spectropolarimeter (Jasco Analytical Instruments, Easton, PA, USA). ESIMS and HRESIMS were acquired on an Agilent 1100 LC/MSD mass spectrometer (Agilent, Santa Clara, CA, USA) and an AB Sciex Triple TOF 5600 spectrometer (AB Sciex Pte. Ltd, Singapore), respectively. 1D and 2D NMR spectra were recorded on Bruker Avance III 400 or 600 MHz spectrometers (Bruker, Fallanden, Switzerland), using the residual solvent signals as the internal standard. All chemical shifts were expressed in ppm. Semi-preparative HPLC was performed on a Waters e2695 system coupled with a 2998 photodiode array (PDA) detector and a 2424 evaporative light-scattering detector (ELSD) (Waters, Milford, MA, USA). A Cosmosil C18 column (5 μM, 10 × 250 mm; flow rate: 3.0 mL/min) and an X-bridge C18 column (5 μM, 10 × 250 mm; flow rate: 3.0 mL/min) were utilized. Thin-layer chromatography (TLC) was performed on pre-coated plates (GF_254_, 0.25 mm, Kang-Bi-Nuo Silysia Chemical Ltd., Yantai, China). TLC spots were visualized under UV light (254 or 365 nm) and by spraying with 5% H_2_SO_4_/vanillin, followed by heating to 120 °C. *Candida albicans* strains (the resistant strain 901, and the sensitive strain 5314) were provided by Dr William A. Fonzi from the Department of Microbiology and Immunology, Georgetown University, Washington DC, USA. RPMI-1640 medium was purchased from Gibco (Life Technologies, Carlsbad, CA, USA). Fluconazole (FLC) was purchased from Pfizer-Roerig Pharmaceuticals (New York, NY, USA).

### 3.2. Plant Material

The unripe fruits of *R. chingii* were purchased from Shanghai Chinese Traditional Medicine Drinking Tablet Co., Ltd., Shanghai, China. They were taxonomically identified by one of the authors (Mr. B. Han). A voucher specimen (No. 20190319) was deposited at the herbarium of the School of Pharmacy at Fudan University.

### 3.3. Extraction and Isolation

The air-dried and powdered unripe fruits (28.0 kg) were extracted with 70% ethanol (5 × 20 L, each time for 24 h) at room temperature. After filtration, the solvent was removed under vacuum to give a dark green residue (5.7 kg, semi-dry), which was suspended in H_2_O (6 L) and then extracted successively with petroleum ether (PE, 3 × 6 L), EtOAc (3 × 6 L), and *n*-BuOH (3 × 6 L). The EtOAc-soluble extract (325.3 g) was subjected to a silica gel column with a stepwise gradient-elution of PE-EtOAc (30:1 → 20:1 → 10:1 → 5:1 → 1:1 → 1:5 → neat EtOAc), to afford nine fractions (Fr.1–9), according to TLC analysis. Fr.3 (10.4 g) was chromatographed over a silica gel column (CH_2_Cl_2_-MeOH, 10:1 → 5:1 → 1:1) to give six subfractions, Fr.3A−F. Compound **4** (3.0 mg) was obtained from Fr.3C (104 mg) by Sephadex LH-20 (MeOH), followed by HPLC purification (MeOH-H_2_O, 70:30, *t*_R_ = 17.5 min). Fr.4 (6.4 g) was fractionated on an MCI column with a step gradient elution of MeOH-H_2_O (50:50 → 70:30 → 85:15 → 100:0), and six fractions (Fr.4A−F) were collected. Separation of Fr.4E (700 mg) over Sephadex LH-20 (MeOH) and semi-preparative HPLC (MeOH-H_2_O, 88:12) afforded compound **7** (28 mg, *t*_R_ = 15.7 min). Fr.5 (23.0 g) was chromatographed over a silica gel column (CH_2_Cl_2_-MeOH, 20:1 → 10:1 → 5:1 → 1:1) to give eight fractions, Fr.5A–H. Compounds **1** (1.0 mg, *t*_R_ = 12.9 min) and **3** (1.0 mg, *t*_R_ = 18.5 min) were obtained from Fr.5A (1.2 g) by Sephadex LH-20 (MeOH), followed by semi-preparative HPLC (MeOH-H_2_O, 70:30). Purification of Fr.5B (400 mg) by semi-preparative HPLC (MeCN-H_2_O, 45:55) yielded compounds **16** (0.8 mg, *t*_R_ = 18.7 min) and **17** (2.5 mg, *t*_R_ = 17.3 min). Fr.5C (600 mg) was further purified by Sephadex LH-20 (MeOH) to give subfractions Fr.5C-1–4. Compounds **18** (1.5 mg, *t*_R_ = 16.0 min) and **19** (2.0 mg, *t*_R_ = 17.4 min) were isolated from Fr.5C-1 (130 mg) by semi-preparative MeCN-H_2_O, 43:57. By employing the same HPLC (MeOH-H_2_O, 67:33) system, Compound **15** (6.0 mg, *t*_R_ = 14.2 min) was purified from Fr.5C-2 (264 mg), whereas compound **20** (2.2 mg, *t*_R_ = 23.5 min) was obtained from Fr.5C-3 (80 mg). Fr.5D (1.2 g) was fractionated by Sephadex LH-20 (MeOH) followed by semi-preparative HPLC (MeCN-H_2_O, 50:50) to furnish compounds **2** (2.0 mg, *t*_R_ = 16.5 min), **8** (5.0 mg, *t*_R_ = 12.3 min), and **14** (6.0 mg, *t*_R_ = 11.2 min). Fr.6 (6.7 g) was fractionated on an MCI column with a step gradient elution of MeOH-H_2_O (30:70 → 50:50 → 70:30 → 85:15 → 100:0), and eight fractions (Fr.6A–H) were collected. Fr.6B (840 mg) was further fractionated by Sephadex LH-20 (MeOH) to give the subfractions Fr.6B-1–3. Fr.6B-2 (260 mg) was purified by semi-preparative HPLC (MeOH-H_2_O, 80:20) to afford compounds **9** (2.0 mg, *t*_R_ = 14.5 min) and **10** (3.3 mg, *t*_R_ = 16.7 min). Fr.6F (0.9 g) was separated over silica gel (CH_2_Cl_2_-MeOH, 20:1 → 10:1 → 5:1 → 1:1) to give four subfractions (Fr.6F-1–4). Compound **5** (25.1 mg, *t*_R_ = 11.3 min) was purified from Fr.6F-1 by Sephadex LH-20 (MeOH), followed by semi-preparative HPLC purification (MeOH-H_2_O, 77:23). Fr.6F-3 was purified by semi-preparative HPLC (MeOH-H_2_O, 74:26) to afford compounds **6** (2.2 mg, *t*_R_ = 21.7 min) and **12** (44.3 mg, *t*_R_ = 20.6 min). Fr.6E (0.9 g) was rechromatographed by silica gel with CH_2_Cl_2_-MeOH (9:1), and six fractions (Fr.6E-1–6) were obtained. Compound **13** (7.0 mg, *t*_R_ = 16.3 min) was isolated from Fr.6E-4 (170 mg) by semi-preparative HPLC (MeOH-H_2_O, 71:29). Fr.7 (4.9 g) was fractionated on an MCI column with a step gradient elution of MeOH-H_2_O (30:70 → 50:50 → 70:30 → 85:15→ 100:0) and seven fractions (Fr.7A−G) were obtained. Separation of Fr.7C (200 mg) over Sephadex LH-20 (MeOH) and semi-preparative HPLC (MeCN-H_2_O, 30:70) afforded compounds **21** (0.7 mg, *t*_R_ = 13.7 min) and **22** (49.1 mg, *t*_R_ = 17.1 min). Fr.7D (550 mg) was purified by semi-preparative HPLC (MeOH-H_2_O, 73:27) to furnish compounds **23** (18.5 mg, *t*_R_ = 9.2 min) and **24** (18.7 mg, *t*_R_ = 11.7 min). Fr.8 (16 g) was fractionated on an MCI column with a step gradient elution of MeOH-H_2_O (30:70 → 50:50 → 70:30 → 85:15 → 100:0) and six fractions (Fr.8A−F) were obtained. Fr.8B (1.09 g) was further separated on a Sephadex LH-20 (MeOH) to give subfractions Fr.8B-1−6. Fr.8B-5 (0.7 g) was purified by semi-preparative HPLC (MeOH-H_2_O, 55:45) to afford compound **25** (5.0 mg, *t*_R_ = 11.2 min). Purification of subfraction Fr.8B-6 (0.8 g) by semi-preparative HPLC (MeCN-H_2_O, 35:65) yielded compound **11** (3.0 mg, *t*_R_ = 22.4 min).

(4*R*,5*R*,8*R*,9*R*,10*R*,14*S*,17*S*,18*S*,19*R*,20*R*)-2,19*α*,23-Trihydroxy-3-oxo-urs-1,12-dien-28-oic acid (rubusacid A, **1**). White powder; [*α*]_D_^21^ 14.0 (*c* 0.2, MeOH); UV (MeOH) *λ*_max_ (log *ε*) 265 (2.31) nm; ECD (*c* 2.67 × 10^−3^ M, MeOH) *λ*_max_ (Δ*ε*): 217 (−16.3), 274 (+13.9), 330 (−3.0) nm; IR (KBr) *v*_max_ 3576, 3446, 2970, 2937, 2870, 1721, 1703, 1690, 1644, 1464, 1402, 1379, 1270, 1242, 1157, 1058, 931, 863, 754 cm^−1^; ^1^H and ^13^C NMR data, see Table 1 and Table 2; ESIMS *m*/*z* 499 [M − H]^−^; HRESIMS *m*/*z* 499.3067 [M − H]^−^ (calcd for C_30_H_43_O_6_, 499.3065, Δ = 0.3 ppm).

(2*R**,4*S**,5*R**,8*R**,9*R**,10*R**,14*S**,17*S**,18*S**,19*R**,20*R**)-2*α*,19*α*,24-Trihydroxy-3-oxo-urs-12-en-28-oic acid (rubusacid B, **2**). White powder; [α]_D_^21^ 11.7 (*c* 0.1, MeOH); UV (MeOH) *λ*_max_ (log *ε*) 202 (2.14) nm; ECD (*c* 1.33 × 10^−3^ M, MeOH) *λ*_max_ (Δ*ε*): 217 (−5.3) nm; IR (KBr) *v*_max_ 3446, 2975, 2927, 2848, 1703, 1691, 1459, 1387, 1210, 1185, 1140, 1053, 1028, 975, 868 cm^−1^; ^1^H and ^13^C NMR data, see Table 1 and Table 2; ESIMS *m*/*z* 501 [M − H]^−^, 525 [M + Na]^+^; HRESIMS *m*/*z* 525.3184 [M + Na]^+^ (calcd for C_30_H_46_O_6_Na, 525.3187, Δ = −0.4 ppm).

(5*R*,8*R*,9*R*,10*R*,14*S*,17*R*,18*S*,19*S*)-2,19*α*-Dihydroxy-olean-1,12-dien-28-oic acid (rubusacid C, **3**). White powder; [α]_D_^21^ 16.3 (*c* 0.1, MeOH); UV (MeOH) *λ*_max_ (log *ε*) 266 (1.53) nm; ECD (*c* 2.75 × 10^−3^ M, MeOH) *λ*_max_ (Δ*ε*): 215 (−20.5), 274 (+12.8), 333 (−4.0) nm; IR (KBr) *v*_max_ 3571, 3446, 2972, 2935, 2868, 1706, 1695, 1634, 1462, 1407, 1384, 1237, 1212, 1157, 1050, 936, 756, 649 cm^−1^; ^1^H and ^13^C NMR data, see Table 1 and Table 2; ESIMS *m*/*z* 485 [M + H]^+^ and 507 [M + Na]^+^; HRESIMS *m*/*z* 485.3254 [M + H]^+^ (calcd for C_30_H_45_O_5_, 485.3262, Δ = −1.6 ppm).

(3*S*,5*S*,8*S*,9*R*,10*S*,13*R*,16*R*)-3*α*,16*α*,17-Trihydroxy-*ent*-kaur-2-one (rubusone, **21**). White powder; [α]_D_^21^ −30.2 (*c* 0.04, MeOH); UV (MeOH) *λ*_max_ (log *ε*) 202 (1.53) nm; ECD (*c* 1.18 × 10^−3^ M, MeOH) *λ*_max_ (Δ*ε*): 285 (−8.7) nm; IR (KBr) *v*_max_ 3379, 2935, 2865, 1701, 1619, 1514, 1449, 1379, 1317, 1290, 1207, 1182, 1142, 1043, 916, 871 cm^−1^; ^1^H and ^13^C NMR data, see Table 1 and Table 3; ESIMS *m*/*z* 337 [M + H]^+^, 359 [M + Na]^+^; HRESIMS *m*/*z* 337.2389 [M + H]^+^ (calcd for C_20_H_33_O_4_, 337.2373, Δ = 4.6 ppm).

### 3.4. NF-κB Inhibitory Assay

HEK293 with a stable NF-*κ*B expression cell line was used for the luciferase assay [11,12,47]. Cells were maintained at 37 °C and 5% CO_2_ atmosphere in Dulbecco’s modified Eagle’s medium with 100 U/mL benzylpenicillin 100 μg/mL streptomycin, 2 mM glutamine, and 10% fetal bovine serum. Before seeding in 96-well plates, the cells were stained for 1 h in serum-free medium supplemented with 2 μ cell Tracker Green CMFDA. Cells were seeded into 96-well plates and incubated for 24 h, and then treated with different concentrations of the tested compounds followed by stimulation with 20 ng/mL TNF-α. The luciferase substrate was added to each well after incubation for 6 h, and then the released luciferin signal was detected using an EnVision microplate reader. The IC_50_ value was derived from a nonlinear regression model (curve-fit), based on a sigmoidal dose response curve (variable slope) and computed using Graphpad Prism 5 (Graphpad Software). Bortezomib (PS-341, CAS No. 179324-69-7) was used as the positive control [11,12,43].

### 3.5. Anti-Fungal Susceptibility Assay

An anti-fungal assay was carried out on the basis of the Clinical and Laboratory Standards Institute (CLSI) method M27-A3 [45,48]. A single colony of *Candida albicans* (the resistant strain 901 or the sensitive strain 5314) was chosen from Sabouraud’s agar and then inoculated in yeast extract peptone dextrose medium (YEPD) for activation by shake bottled cultivation (200 rpm, 37 °C). After 16 h, fungi in the late-exponential growth phase were obtained, followed by being adjusted to 1 × 10^3^–5 × 10^3^ CFU/mL with RPMI 1640 medium. The density of the fungi in 96-well plates was 10^3^ CFU/mL, while the final concentrations of the test compounds ranged from 64 to 0.125 μg/mL in triplicate. The assay volume in each well was 100 μL with MIC_80_ determined following 48 h incubation at 37 °C. Optical density was measured with a microplate reader (Multiskan MK3; Labsystems, Nantaa, Finland) at 630 nm. MIC_80_ was determined as the lowest concentration of the drugs that inhibited growth by 80% compared with the positive control. Fluconazole was used as the positive control [45,46].

## 4. Conclusions

Previous phytochemical studies on the fruits and leaves of *R. chingii* were extensively reviewed [2,5]. Around 15 triterpenoids and 17 diterpenoids have so far been obtained from *R. chingii* [2,5,49]. In the present work, we focused on triterpenoids and diterpenoids from the unripe fruits of *R. chingii*. Three highly oxygenated triterpenoids (rubusacids A-C, **1**–**3**) and one *ent*-kaurane-type diterpenoid (rubusone, **21**) were reported that were hitherto unknown. Compounds **1** and **3** possess a special diosphenol unit in ring A. Diosphenols are *α*-diketones, in which one of the carbonyls is enolized; they are part of the structure of several products of natural and synthetic origin, but are quite rare in naturally occurring ursane- and oleanane-type pentacyclic triterpenoids. To our knowledge, only a few plant-originated oleanane-/ursane-type [13,50,51,52,53] and one biotransformed ursane-type [54] triterpenoids have such a moiety in ring A. These cucurbitane-type tetracyclic triterpenoids (**15**–**20**) were obtained from the unripe fruits of *R. chingii* for the first time. Regarding the bioactivity evaluations, three isolates (**15**–**17**) showed potent inhibitory effects against NF-*κ*B. It is worth noting that some cucurbitane-type tetracyclic triterpenoid glycosides from *Momordica charantia* [55] have also been found to have this kind of effect. The major component, 2α,3α,23-trihydroxy-urs-12-en-28-oic acid (**12**), might be the principle in the anti-fungal activity of 70% ethanol Fructus Rubi [45]. In general, the above findings expanded the terpenoic structure diversity of *R. chingii* and could provide useful clues for the discovery and development of new therapeutic or preventive agents for the treatment of NF-*κ*B related diseases.

## Figures and Tables

**Figure 1 molecules-26-01911-f001:**
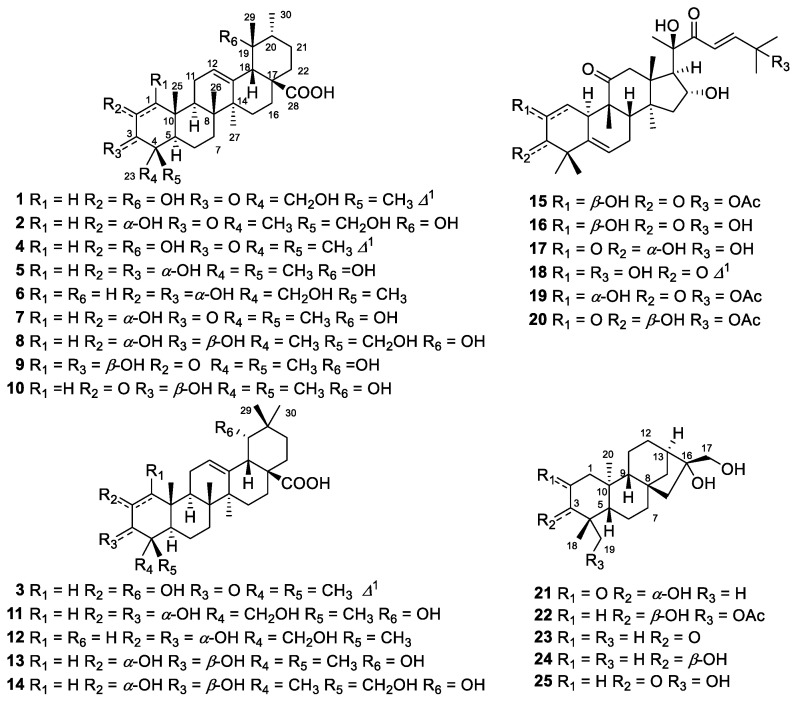
Chemical structure of triterpenoids (**1**–**20**) and diterpenoids (**21**–**25**).

**Figure 2 molecules-26-01911-f002:**
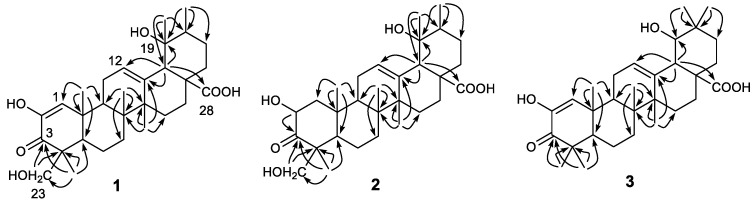
Observed key HMBC correlations of triterpenoids **1**–**3**.

**Figure 3 molecules-26-01911-f003:**
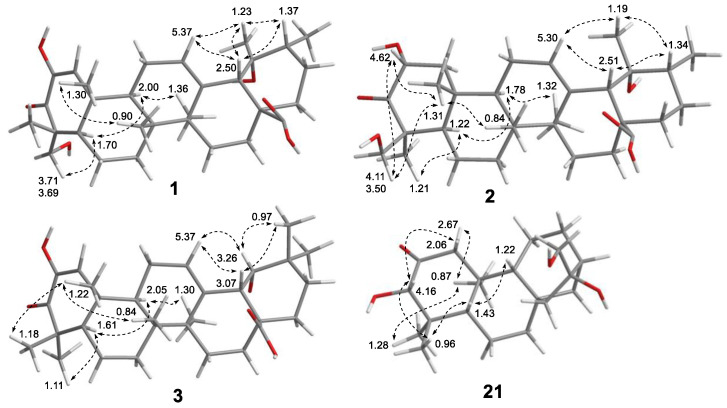
Observed key ROE correlations of compounds **1**–**3** and **21**.

**Table 1 molecules-26-01911-t001:** ^13^C NMR data *^a^* (*δ* in ppm, 150 MHz) of compounds **1**–**3** (in CD_3_OD) and **21** (in C_6_D_6_).

No.	1	2	3	21	No.	1	2	3	21
1	129.5	51.2	130.0	53.0	16	26.6	26.6	29.4	81.2
2	145.8	71.2	146.0	210.2	17	48.7	49.0	46.7	66.3
3	200.5	215.4	202.4	82.8	18	55.2	55.1	45.3	29.6
4	51.6	56.4	45.5	45.6	19	73.5	73.6	82.4	16.6
5	55.8	60.2	55.2	54.0	20	43.1	43.1	36.1	18.4
6	19.9	20.9	20.2	20.4	21	27.3	27.3	28.6	
7	34.6	34.4	33.7	41.5	22	38.9	38.9	34.0	
8	41.7	41.2	41.3	45.1	23	65.9	20.1	22.2	
9	44.6	48.5	44.7	55.6	24	22.0	66.0	28.0	
10	39.4	39.0	39.7	45.2	25	20.5	17.6	20.0	
11	24.9	24.8	24.3	18.8	26	17.8	17.3	18.0	
12	129.0	128.9	124.5	26.1	27	24.8	25.0	25.1	
13	140.4	140.2	145.0	44.6	28	182.2	182.2	182.3	
14	42.9	42.7	43.0	36.8	29	27.0	27.1	28.1	
15	29.5	29.6	29.5	52.9	30	16.6	16.6	25.1	

*^a^* Assignments were made by a combination of 1D and 2D NMR experiments.

**Table 2 molecules-26-01911-t002:** ^1^H NMR data *^a^* (*δ* in ppm, *J* values in Hz, 600 MHz) of **1**–**3** in CD_3_OD.

No.	1	2	3
1*α*	6.29 s	1.17 dd (12.7, 12.5)	6.27 s
1*β*		2.31 dd (12.5, 6.6)	
2		4.62 dd (12.7, 6.6)	
5	1.70 dd (overlapped)	1.22 dd (overlapped)	1.61 m
6a	1.66 m	1.63 m	1.60 m
6b	1.62 m	1.58 m	1.58 m
7a	1.76 m	1.55 m	1.76 m
7b	1.63 m	1.35 m	1.58 m
9	2.00 dd (11.3, 6.3)	1.78 m	2.05 dd (10.7, 6.6)
11a	2.27 ddd (17.3, 11.3, 3.5)	2.06 m	2.33 m
11b	2.14 ddd (17.3, 6.3, 3.7)	1.35 m	2.19 m
12	5.37 dd (3.7, 3.5)	5.30 dd (3.8, 3.3)	5.37 dd (3.8, 3.1)
15*α*	1.03 ddd (14.1, 4.2, 2.5)	0.99 m	1.02 m
15*β*	1.83 ddd (14.1, 13.2, 4.4)	1.81 m	1.63 m
16*α*	2.60 ddd (13.3, 13.2, 4.4)	2.57 ddd (13.4, 13.4, 4.0)	1.75 m
16*β*	1.55 ddd (13.3, 4.2, 2.5)	1.55 m	1.60 m
18	2.50 s	2.51 s	3.07 d (3.9)
19			3.26 d (3.9)
20	1.37 m	1.34 m	
21a	1.25 m	1.22 m	2.29 m
21b	1.20 m	1.16 m	1.62 m
22a	1.75 m	1.73 m	1.61 m
22b	1.62 m	1.62 m	1.37 m
23a	3.71 d (12.0)	1.21 s	1.11 s
23b	3.69 d (12.0)		
24a	1.29 s	4.11 d (11.4)	1.18 s
24b		3.50 d (11.4)	
25	1.30 s	1.31 s	1.22 s
26	0.90 s	0.84 s	0.84 s
27	1.36 s	1.32 s	1.30 s
29	1.23 s	1.19 s	0.97 s
30	0.95 d (7.1)	0.93 d (6.3)	0.94 s

*^a^* Assignments were made by a combination of 1D and 2D NMR experiment.

**Table 3 molecules-26-01911-t003:** ^1^H (*δ* in ppm, 600 MHz) and ^13^C (*δ* in ppm, 150 MHz) NMR data *^a^* of **21**.

No.	21		No.	21	
	*δ*_H_ (*J* in Hz) *^b^*	*δ*_H_ (*J* in Hz) *^c^*		*δ*_H_ (*J* in Hz) *^b^*	*δ*_H_ (*J* in Hz) *^c^*
1	1.37 d (12.4)	2.06 d (12.2)	12	1.62 m; 1.65 m	1.87 m; 1.83 m
	2.41 d (12.4)	2.67 d (12.2)	13	1.88 m	2.46 m
3	3.62 d (3.8)	4.16 s	14	1.57 m; 0.69 m	1.70 m; 0.85 m
5	0.91 br d (11.6)	1.43 br d (11.5)	15	1.39 m; 1.25 m	1.63 m; 1.35 m
6	1.32 m; 1.05 m	1.69 m; 1.37 m	17	3.44 d (10.3)	4.12 d (11.0)
7	1.22 m; 1.21 m	1.56 m; 1.53 m		3.37 d (10.3)	4.05 d (11.0)
9	0.99 br d (6.7)	1.22 br d (8.6)	18	1.10 s	0.96 s
11	1.62 m; 1.18 m	1.84 m; 1.51 m	19	0.67 s	1.28 s
OH-3	3.72 d (3.8)		20	0.69 s	0.87 s

*^a^* Assignments were made by a combination of 1D and 2D NMR experiments; *^b^* Measured in C_6_D_6_;*^c^* Measured in C_5_D_5_N.

**Table 4 molecules-26-01911-t004:** Inhibitory activities of indicated compounds against NF-*κ*B.

Compound	NF-*κ*B (IC_50_) *^a^*
**15**	0.08 ± 0.03 μM
**16**	0.61 ± 0.12 μM
**17**	1.60 ± 0.32 μM
PS-341 *^b^*	0.44 ± 0.08 μM

*^a^* These data are expressed as the mean SEM of triplicated experiments. *^b^* Positive control.

## Data Availability

All data and figures generated or used during the study appear in the submitted article.

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
