# Peer review of "Highly Oxygenated Triterpenoids and Diterpenoids from Fructus Rubi (*Rubus chingii* Hu) and Their NF-kappa B Inhibitory Effects"

_molecules, 2021, doi:10.3390/molecules26071911_

Round 1

Reviewer 1 Report

The authors in their research article present results related to the isolation of triterpenoids and diterpenoids from the fruit of Rubus chingii. Among the isolated compounds, three highly oxygenated triterpenoids and one ent-kaurane-type diterpenoid were reported as previously unidentified. The compounds were evaluated for their ability to inhibit NF-κB activity  and three cucurbitane-type compounds were reported with such activity. The authors also examined the antifungal effects of the isolated compounds towards Candida albicans, but identified minimal inhibitory activity of only one of the isolates.

Overall, the manuscript presents some novel findings regarding the chemical composition of  Rubus chingii and NF-κB inhibitory activity of Rubus-derived compounds. I have the following questions and comments for the authors:

The research regarding the antifungal activity is too preliminary to conclude that the findings could provide a starting point for the development of antifungal agents. If the authors want to focus on the analysis of the antifungal activity, I would recommend analyzing a wider panel of fungi species. The authors followed up in their study on a previous investigation of the activity of the Rubus extract. The extract only potentiated the activity of fluconazole with no activity of the extract alone, thus it would be more appropriate to examine the compounds in combination with fluconazole.

There is a lack of coherency in the research aims of this study. What is the association between the analysis of the NF-κB inhibitory activity and anti-fungal activity of the examined compounds? Based on the obtained results it would be more coherent with the obtained NF-κB-inhibitory activity to perform bioactivity analysis toward cell lines instead of fungi. I would suggest that the authors focus their research article on only one aspect of the activity of the isolated compounds.  Since the authors showed minor activity of compounds towards Candida,  I would recommend focusing on the activity of compounds  as mediators of NF-kB inhibition.

The manuscript requires some language corrections. Below are a few provided modifications:

Line 43: as a herb tonic

Line 48: encountered in the fruits

Line 55: a commercially available

Line 194: The rest of the isolates were inactive

Line 338: Around 15 triterpenoids and 17 diterpenoids have been so  far obtained from R. chingii

Author Response

Many thanks for your  comments and suggestions on our manuscript. Please see the attachment for our point-by-point responses to the comments and suggestions.

Reviewer 2 Report

Isolation and identification are challenging in natural product chemistry. The authors reinvestigated on the unripe fruits of Rubus chingii Hu and identified four natural products that haven't been reported. The whole article was also centered on the characterization of those four natural products. The rest of the article is redundant. These superfluous things make the article neither fish nor fowl. If the authors focus on those four compounds, it will be better. Suggest the authors reorganize their manuscript and think about the title carefully. If it's necessary, a 'Communication' is better than an 'Article'.

Fufenzi acid or Fufenzic acid?The authors used both expressions. Please double-check. 

Table 4 reported known compounds' NF-κB inhibitory and antifungal activity. Unfortunately, I didn't see the data of those four new isolated compounds in table 4. In the results and discussion section, the reason why the authors chose NF-κB Inhibitory Assay for known compounds is not really clear, although they mentioned something in the manuscript. 

The NF-κB Inhibitory Assay experiment protocol is not clear. Suggest providing full protocol including cell culture protocol. 

Author Response

(The authors gave the same response as above.)

Round 2

Reviewer 1 Report

The authors have addressed all my comments and have made the recommended modifications, thus I find the manuscript appropriate for publication. 

Reviewer 2 Report

No more concerns.